# RATIO-RESIDUAL DIFFUSION MODEL FOR IMAGE RESTORATION

## ABSTRACT

Most existing diffusion-based image restoration methods suffer from poor interpretability and inefficient sampling, due to their direct incorporation of degraded images as conditions within the original diffusion models. Recently, some researches have tried to build a new diffusion model by transferring the discrepancies between degraded and clear images, however, they cannot effectively model diverse degradation. To address these issues, we propose a universal diffusion model for image restoration that can cover different types of degradation. Specifically, our method consists of a Markov chain that convert a high-quality image to its low-quality counterpart. The transition kernel of this Markov chain is constructed through the ratio and residual between the high-quality and low-quality images, which provides a general expression that can effectively handle various degradation processes. Moreover, we analyze the characteristics of different degradation, and design an exponential schedule that enables flexible control over the diffusion speed pertaining to different degradation, which yields better restoration performance. Extensive experiments demonstrate that our method achieves superior or at least comparable performance compared with existing image restoration methods on multiple image restoration tasks, including low-light image enhancement, deraining, deblurring, denoising, and dehazing.

## 1 INTRODUCTION

Image restoration aims to recover the high-quality image from its degraded low-quality counterpart, which is challenging and considered severely ill-posed due to the infinite potential solutions for a given degraded image. With the development of deep learning, convolution neural network (CNN)-based methods have dominated the image restoration tasks, including low-light image enhancement (Wu et al., 2022), deraining (Ren et al., 2019), deblurring (Ren et al., 2021), denoising (Chang et al., 2020), dehazing (Yu et al., 2022), and numerous others. Recently, diffusion models (Ho et al., 2020; Song et al., 2020a;b) have achieved great progress in image generation (Dhariwal & Nichol, 2021), showcasing promising performance across various downstream tasks, such as image editing (Hertz et al., 2022) and personalization (Ruiz et al., 2023). Current research is exploring the potential of powerful diffusion models in addressing the challenging image restoration tasks.

There are two popular applications of diffusion models to image restoration. The first one is directly training a diffusion model that is conditioned on the degraded images (Saharia et al., 2022; Yi et al., 2023). These approaches diffuse the high-quality images into a pure Gaussian white noise in the forward process, and then utilize the degraded image as a condition to recover the clean image during the reverse process. Such heuristic training procedure lacks interpretability, and typically suffers from unstable restoration (Jiang et al., 2023). The second one is to use a pre-trained diffusion model (e.g., DDPM (Ho et al., 2020)) as a prior to constraint the restoration process (Feng et al., 2023; Mardani et al., 2023; Wang et al., 2023a; Fei et al., 2023). These methods are commonly based on Bayesian inference, which take the degradation model as the condition to constraint the sampling of the pre-trained diffusion model in the reverse process. However, most of them require the knowledge of the degradation model, which limit their applicability in complex real-world scenarios. On the other hand, the direct inheritance of the fundamental logic from the pre-trained DDPM model may result in inefficient sampling.

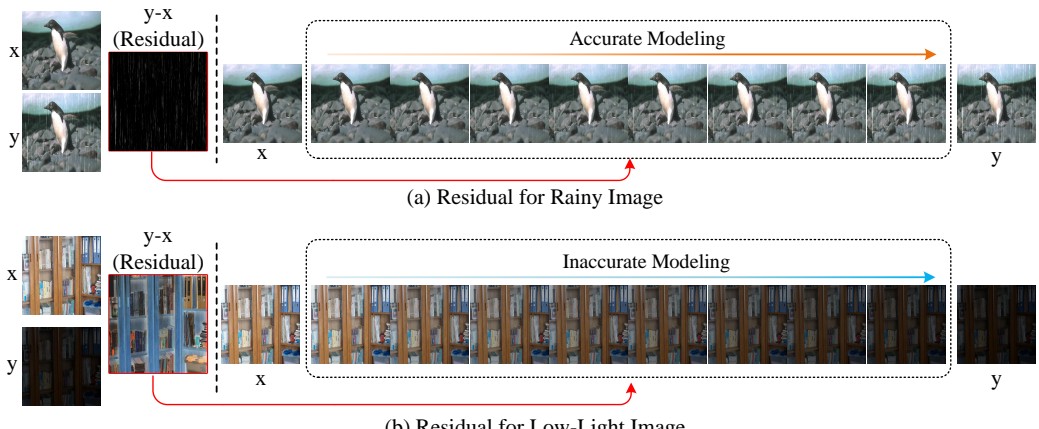

Figure 1: Illustration of residual-based diffusion when it comes to different degradation. Note that we ignore the noise term in diffusion model for better observation. (a) For deraining (the degradation operator $\mathbf{A} = \mathbf{E}$), the residual-based diffusion can model the degradation process well. (b) For low-light image enhancement ($\mathbf{A} \neq \mathbf{E}$), it fails to accurately model the degradation process.

It is imperative to develop a universal and adaptable diffusion model tailored for image restoration. Recent works (Yue et al., 2023) have attempted to solve this problem, which diffuse a high-quality image to its degraded version by shifting the residual between them. While promising, these methods still face a critical issue. The direct residual-based shifting approach exhibits limited adaptability when applied to different image restoration tasks. Image degradation is commonly formulated as $\mathbf{y} = \mathbf{Ax} + \mathbf{n}$, where $\mathbf{y}$, $\mathbf{x}$, $\mathbf{A}$, and $\mathbf{n}$ denote the degradation observation, the latent clear image, degradation operator, and noise, respectively. The residual-based methods typically model the difference $(\mathbf{y} - \mathbf{x}) = (\mathbf{A} - \mathbf{E})\mathbf{x} + \mathbf{n}$, where $\mathbf{E}$ is the all-ones matrix. When $\mathbf{A} = \mathbf{E}$, the degradation is primary caused by $\mathbf{n}$, which can be represented well by the residuals $(\mathbf{y} - \mathbf{x})$, as shown in Fig. 1(a). However, when the degradation operator $\mathbf{A}$ significantly deviates from $\mathbf{E}$, the residual-based methods struggle to accurately represent the degradation component $\mathbf{A}$, and cannot model the degradation process effectively, resulting in unsatisfactory restoration performance, as shown in Fig. 1(b). Therefore, residual-based diffusion is not universally suitable for all types of degradation.

To address the aforementioned issue, in this paper, we propose a universal diffusion model that can be effectively adapted to various image restoration tasks, named R2Diff. Specifically, we design a novel Markov process that encompasses the transition from the initial state, representing the distribution of clear images, to the final state, representing the distribution of the corresponding degraded images. We represent degradation process through the ratio and residual between the clean image $\mathbf{x}$ and the degraded image $\mathbf{y}$, and design a new Markov transition kernel based on this. Compared with residual-based diffusion, our method allows for a more accurate modeling of a wider range of degradation processes. Moreover, we revisit the properties of different degradation, and develop an exponential schedule that allows flexible control over the diffusion speed for different degradation. It can further enhance the adaptability of our diffusion model to diverse degradation.

We summarize our main contributions as follows:

- We propose a universal diffusion model tailored for image restoration tasks. It consists of a Markov chain that transfers the clear image to its degraded version, whose transition kernel is based on the ratio and residual between them. We further develop an exponential schedule to flexibly control the diffusion speed for different degradation.
- Extensive experiments on multiple image restoration tasks demonstrate the effectiveness of the proposed diffusion model, including low-light image enhancement, deraining, deblurring, denoising, and dehazing.

## 2 BACKGROUND

We follow the work Denoising Diffusion Probabilistic Models (DDPM) (Ho et al., 2020), and briefly introduce the key concepts underlying it. It consists of two processes: a $T$-steps forward process

that gradually adds noise to the input image, and a reverse process that learns to generate images by iterative denoising over the same $T$ steps. For any time $t \in [0, T]$, we can get the current state $\mathbf{x}_t$ through the forward process formulated by:

$$q(\mathbf{x}_t|\mathbf{x}_{t-1}) = \mathcal{N}(\mathbf{x}_t; \sqrt{1-\beta_t}\mathbf{x}_{t-1}, \beta_t\mathbf{I}), \tag{1}$$

where $\mathbf{x}_t$ is the noisy image at $t$, $\beta_t$ is the coefficient that determines the variance added in each iteration, and $\mathbf{I}$ is the identity matrix. So, we can get the $\mathbf{x}_t$ with the forward process:

$$\mathbf{x}_t = \sqrt{1-\beta_t}\mathbf{x}_{t-1} + \beta_t\epsilon, \quad \epsilon \sim \mathcal{N}(\mathbf{0}, \mathbf{I}) \tag{2}$$

Through the reparameterization, we can get the $\mathbf{x}_t$ given the starting state $\mathbf{x}_0$:

$$q(\mathbf{x}_t|\mathbf{x}_0) = \mathcal{N}(\mathbf{x}_t; \sqrt{\bar{\alpha}_t}\mathbf{x}_0, (1-\bar{\alpha}_t)\mathbf{I}), \quad \bar{\alpha}_t = 1 - \beta_t, \quad \bar{\alpha}_t = \prod_{s=1}^{t}\alpha_t \tag{3}$$

The reverse process aims to estimate the previous state $\mathbf{x}_{t-1}$ given the current state $\mathbf{x}_t$. We can obtain the posterior distribution $p(\mathbf{x}_{t-1}|\mathbf{x}_t, \mathbf{x}_0)$ through the Bayes' theorem:

$$p(\mathbf{x}_{t-1}|\mathbf{x}_t, \mathbf{x}_0) = \mathcal{N}(\mathbf{x}_{t-1}; \boldsymbol{\mu}_t(\mathbf{x}_t, \mathbf{x}_0), \sigma_t^2\mathbf{I}), \tag{4}$$

$$\text{where} \quad \boldsymbol{\mu}_t(\mathbf{x}_t, \mathbf{x}_0) = \frac{1}{\sqrt{\alpha_t}}(\mathbf{x}_t - \frac{1-\alpha_t}{\sqrt{1-\bar{\alpha}_t}}\epsilon), \quad \sigma_t^2 = \frac{1-\bar{\alpha}_{t-1}}{1-\bar{\alpha}_t}\beta_t, \tag{5}$$

DDPM leverages a neural network $\epsilon_\theta$ to estimate the noise term $\epsilon$ in Eq. 5. For any time-step $t \in [0, T]$, we can get the loss function defined in (Ho et al., 2020):

$$L(\theta) = \mathbb{E}_{\mathbf{x}_0 \sim q(\mathbf{x}_0), \epsilon \sim \mathcal{N}(\mathbf{0}, \mathbf{I})} \left[ \| \epsilon - \epsilon_\theta \left( \sqrt{\bar{\alpha}_t}\mathbf{x}_0 + \sqrt{1-\bar{\alpha}_t}\epsilon \right) \|_2^2 \right]. \tag{6}$$

In the reverse process, through the iterative sampling of $\mathbf{x}_{t-1}$ from the posterior distribution, DDPM can generate a sample $\mathbf{x}_0 \sim q(x_0)$ from a pure Gaussian noise $\mathbf{x}_T \sim \mathcal{N}(\mathbf{0}, \mathbf{I})$. Here, $q(x_0)$ denotes the data distribution of the training dataset.

## 3 METHOD

In this section, we will give a detailed introduction to the proposed R2Diff, a diffusion model tailored for various image restoration tasks. It consists of a Markov chain that transfers the clear image to its degraded counterpart by using the ratio and residual between them, which will be detailed in Sec. 3.1. Moreover, an exponential schedule is designed to flexibly control the diffusion speed for different types of degradation, which will be described in Sec. 3.2.

### 3.1 DIFFUSION PROCESS OF THE R2DIFF

Many works have proven the effectiveness of the iterative paradigm of diffusion on specific image restoration tasks, which inspire us to explore a universal diffusion model capable of effectively adapting to diverse image restoration tasks. In this section, we present the new Markov chain designed for various image restoration tasks, and introduce the forward and reverse process in details.

**Forward Process.** Existing works (Luo et al., 2023; Yue et al., 2023) have attempted to model the degradation process between clear image and degraded image through directly shifting their residuals, which we have proven ineffective for some degradation (see Fig. 1). Let us revisit the general degradation process:

$$\mathbf{y} = \mathbf{A}\mathbf{x_0} + \mathbf{n} \tag{7}$$

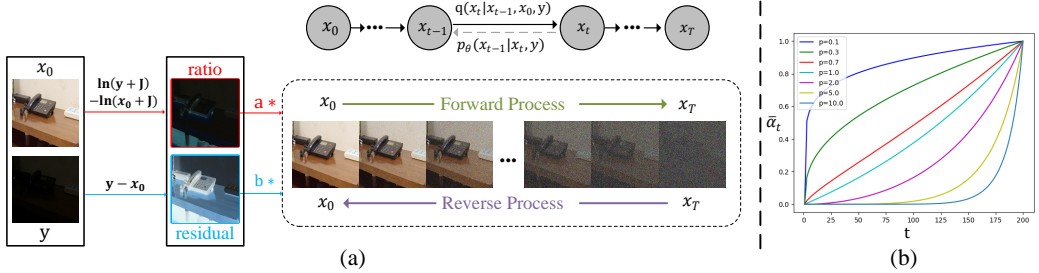

(a)                                                                    (b)

Figure 2: Framework of the proposed R2Diff. (a) The forward and reverse process of the R2Diff. During the forward process, the clear image $\mathbf{x}_0$ gradually diffuses into its degraded counterpart with noise, i.e., $\mathbf{x}_T \in \mathcal{N}(\mathbf{x_T}; \mathbf{y}, \bar{\beta}_t \mathbf{I})$, through transferring the ratio and residual between $\mathbf{x}_0$ and $\mathbf{y}$. In the reverse process, the degradation is progressively removed from $\mathbf{x}_T$, accompanied by the elimination of noise, restoring the degraded image to its clear state. (b) Exponential schedule: the value of $\bar{\alpha}_t$ with time-step $t$ under different settings of $p$.

where $\mathbf{x}_0$ is the clear image, $\mathbf{y}$ denotes the corresponding degraded image. It contains two key components, the noise term $\mathbf{n}$ and the degradation operator $\mathbf{A}$. The former can be captured by residuals between $\mathbf{y}$ and $\mathbf{x}_0$ (Yue et al., 2023), while the latter cannot be represented effectively through the same approach. Inspired by the works (Wang et al., 2022; Fu et al., 2021), we represent $\mathbf{A}$ through the ratio, i.e., $\mathbf{A} = \mathbf{y}/\mathbf{x_0}$.

However, the direct ratio form, i.e., $\mathbf{A} = \mathbf{y}/\mathbf{x_0}$, is difficult to model into the transition kernel of the Markov process. To remedy this issue, we adopt the logarithmic function and get the transfers:

$$\mathbf{r_0} = a \underbrace{[\ln(\mathbf{y} + \mathbf{E}) - \ln(\mathbf{x_0} + \mathbf{E})]}_{\text{represent } \mathbf{A}} + b \underbrace{(\mathbf{y} - \mathbf{x_0})}_{\text{represent } \mathbf{n}}, \tag{8}$$

where $\mathbf{r_0}$ denotes the transition between $\mathbf{x_0}$ and $\mathbf{y}$, $\mathbf{E}$ represents the all-ones matrix, $a$ and $b$ are the weighting coefficients, where $a + b = 1$. We leverage the logarithm of the ratio between $\mathbf{y}$ and $\mathbf{x}_0$ to represent $\mathbf{A}$, and use the residual between $\mathbf{y}$ and $\mathbf{x}_0$ to represent $\mathbf{n}$. Beyond the residual-based modeling, our method takes both degradation operator $\mathbf{A}$ and noise term $\mathbf{n}$ into consideration, which can cover a wide range of degradation process. More importantly, Eq. 8 also satisfies that $\mathbf{y} = \mathbf{x_0} + \mathbf{r_0}$ when $t = T$. The proof is provided in Appendix A.1. Note that $a$ in Eq. 8 determines the influence of ratio and residual on $\mathbf{r}_0$. Therefore, it is necessary to adjust the value of $a$ flexibly for different degradation. The choice of $a$ for different degradation will be discussed in Sec. 5.

After that, we construct a Markov chain that transfers from $\mathbf{x_0}$ to $\mathbf{y}$ through shifting $\mathbf{r_0}$. The forward process can be formulated as:

$$q(\mathbf{x}_{1:T}|\mathbf{x}_0, \mathbf{y}) = \prod_{t=1}^{T} q(\mathbf{x}_t|\mathbf{x}_{t-1}, \mathbf{y}), \quad q(\mathbf{x}_t|\mathbf{x}_{t-1}, \mathbf{y}) = \mathcal{N}(\mathbf{x}_t; \mathbf{x}_{t-1} + \alpha_t \mathbf{r}_0, \beta_t \mathbf{I}), \tag{9}$$

where $\mathbf{I}$ is the identity matrix, the two sets of hyper-parameters $\alpha_t$ and $\beta_t$ control the diffusion speed of $\mathbf{r}_0$ and noise, respectively. $\alpha_t > 0$, $\beta_t > 0$, and $t = 1, 2, ..., T$. The selection of $\alpha_t$ and $\beta_t$ will be detailed in Sec. 3.2. Then, we can get the marginal probability distributions:

$$q(\mathbf{x}_t|\mathbf{x}_0, \mathbf{y}) = \mathcal{N}(\mathbf{x}_t; \mathbf{x}_0 + \bar{\alpha}_t \mathbf{r}_0, \bar{\beta}_t \mathbf{I}) \tag{10}$$

where $\bar{\alpha}_t = \sum_{i=1}^{t} \alpha_i$, $\bar{\beta}_t = \sum_{i=1}^{t} \beta_i$. The $\bar{\alpha}_t$ increases monotonically as time $t$ increases. If $t = 0$, $\bar{\alpha}_t \to 0$, and if $t = T$, $\bar{\alpha}_t \to 1$.

**Reverse Process.** In the reverse process, we try to yield the clear image $\mathbf{x}_0$ from the final state through iterative sampling step by step, which is formulated by:

$$p_\theta(\mathbf{x}_0|y) = \int p_\theta(\mathbf{x}_{0:T}|y)d\mathbf{x}_{1:T}, \quad p_\theta(\mathbf{x}_{0:T}|y) = p(\mathbf{x}_T|y)\prod_{t=1}^{T} p_\theta(\mathbf{x}_{t-1}|\mathbf{x}_t, y), \tag{11}$$

where $p(\mathbf{x}_T|y) = \mathcal{N}(\mathbf{x}_T; \mathbf{y}, \bar{\beta}_t\mathbf{I})$ denotes the final state when $t = T$, and the $p_\theta(\mathbf{x}_{t-1}|\mathbf{x}_t, y) = \mathcal{N}(\mathbf{x}_{t-1}; \boldsymbol{\mu_\theta}(\mathbf{x}_t, \mathbf{y}, t), \boldsymbol{\Sigma_\theta}(\mathbf{x}_t, \mathbf{y}, t))$ represents the reverse transfer from current state $\mathbf{x}_t$ to previous state $\mathbf{x}_{t-1}$ with learnable parameters $\theta$. Following (Ho et al., 2020), we train the parameter $\theta$ by optimizing the usual variational bound on negative log likelihood, which is:

$$L(\theta) = \sum_{t>0} D_{KL}\left[q(\mathbf{x}_{t-1}|\mathbf{x}_t, \mathbf{x}_0, \mathbf{y})||p_\theta(\mathbf{x}_{t-1}|\mathbf{x}_t, \mathbf{y})\right], \qquad (12)$$

where $D_{KL}$ is the KL divergence. The reverse transfer probability $q(\mathbf{x}_{t-1}|\mathbf{x}_t, \mathbf{x}_0, \mathbf{y})$ can be represented through the Bayes' rule:

$$q(\mathbf{x}_{t-1}|\mathbf{x}_t, \mathbf{x}_0, \mathbf{y}) = \mathcal{N}(\mathbf{x}_{t-1}; \tilde{\boldsymbol{\mu}}_t(\mathbf{x}_t, \mathbf{x}_0, \mathbf{y}), \tilde{\beta}_t\mathbf{I}), \qquad (13)$$

where $\tilde{\boldsymbol{\mu}}_t(\mathbf{x}_t, \mathbf{x}_0, \mathbf{y})$ and $\tilde{\beta}_t\mathbf{I}$ are formulated by:

$$\tilde{\boldsymbol{\mu}}_t(\mathbf{x}_t, \mathbf{x}_0, \mathbf{y}) = \frac{\bar{\beta}_{t-1}}{\bar{\beta}_t}\mathbf{x}_t + \frac{\mathbf{K}_t}{\bar{\beta}_t}[a\ln(\mathbf{y}+\mathbf{E}) + b\mathbf{y}] + \frac{1}{\bar{\beta}_t}(\beta_t\mathbf{x}_0 - \mathbf{K}_t[a\ln(\mathbf{x}_0+\mathbf{E}) + b\mathbf{x}_0]),$$
$$\tilde{\beta}_t = \frac{\beta_t\bar{\beta}_{t-1}}{\bar{\beta}_t}, \qquad (14)$$

In Eq. 14, $\mathbf{K}_t = \beta_t\bar{\alpha}_{t-1} - \bar{\beta}_{t-1}\alpha_t$. The detailed derivation is provided in Appendix A.2. Following DDPM (Ho et al., 2020), we set $\boldsymbol{\Sigma_\theta}(\mathbf{x}_t, \mathbf{y}, t) = \tilde{\beta}_t\mathbf{I}$ to untrained time dependent constants since $\tilde{\beta}_t$ is unrelated to $\mathbf{x}_t$ and $\mathbf{y}$. With $p_\theta(\mathbf{x}_{t-1}|\mathbf{x}_t, y) = \mathcal{N}(\mathbf{x}_{t-1}; \boldsymbol{\mu_\theta}(\mathbf{x}_t, \mathbf{y}, t), \tilde{\beta}_t\mathbf{I})$, we can simplify the loss function in Eq. 12 through reparameterization, which is:

$$L(\theta) = \sum_{t>0} \frac{1}{2\tilde{\beta}_t\bar{\beta}_t^2}\|\boldsymbol{\gamma}_t[f_\theta(\mathbf{x}_t, \mathbf{y}, t) - \mathbf{x}_0] - \boldsymbol{\lambda}_t[\ln(f_\theta(\mathbf{x}_t, \mathbf{y}, t) + \mathbf{E}) - \ln(\mathbf{x}_0 + \mathbf{E})]\|_2^2 \qquad (15)$$

where $f_\theta$ is a network whose parameter is $\theta$, $\boldsymbol{\gamma}_t$ and $\boldsymbol{\lambda}_t$ are time-dependent hyper-parameters. The detailed derivation is provided in Appendix A.3. Fig 2(a) shows the whole forward and reverse process of our diffusion model.

## 3.2 EXPONENTIAL SCHEDULE

In Eq. 10, the coefficient $\bar{\alpha}_t$ and $\bar{\beta}_t$ control the diffusion speed of degradation and noise, respectively. Inspired by (Liu et al., 2023), we employ two independent coefficient schedules to $\bar{\alpha}_t$ and $\bar{\beta}_t$ in the R2Diff, which allows us to explore the schedules of $\bar{\alpha}_t$ and $\bar{\beta}_t$ more flexibly. During the reverse process of diffusion, denoising is accompanied by the removal of degradation. To effectively address various degradation, it is essential to use different coefficient sets for $\bar{\alpha}_t$ and $\bar{\beta}_t$ that can adapt to specific degradation characteristics, instead of using a fixed set.

In our work, we focus on $\bar{\alpha}_t$, since it controls the diffusion speed of degradation in the forward process, and regulates the speed of degradation removal in the reverse process. Many works (Rissanen et al., 2022; Choi et al., 2022) have proven that diffusion models perform a coarse-to-fine manner in the reverse process, which means they first recover the global contents of images when the time-step $t$ is large, and learn fine-grained details when $t$ is small. As the noise is progressively removed, the diffusion's focus gradually shifts from the restoration of global contents to the enhancement of fine-grained details. Intuitively, for those degradation that severely damages the global contents, the $\bar{\alpha}_t$ should have a higher changing rate when $t$ is large (see Fig. 6), which allows the diffusion model to fully perceive the degradation when $t$ is large. In the reverse process, it will effectively eliminate the degradation when recovering global contents. Conversely, for those degradation that mainly ruins the fine-grained details of images, the $\bar{\alpha}_t$ should have a higher changing rate when $t$ is small.

Based on the aforementioned analysis, we design an exponential schedule for $\bar{\alpha}_t$. By adjusting $\bar{\alpha}_t$, we can flexibly control the diffusion speed for different degradation, thereby facilitating the degradation removal in the reverse process. Specifically, we design our schedule for $\bar{\alpha}_t$ as:

$$\bar{\alpha}_t = \frac{1}{e-1} * \left[ exp[(\frac{t}{T})^p] - 1 \right], \quad t = 1, 2, 3, ..., T \tag{16}$$

where $p$ controls changing rate of $\bar{\alpha}_t$. Through adjusting $p$, we can flexibly control the diffusion speed for different degradation, as shown in Fig. 2(b). The selection of $p$ for different degradation will be described in Sec. 5.

As for $\bar{\beta}_t$, we explore different schedules and find that the commonly-used cosine schedule in DDPM (Ho et al., 2020) works well for different degradation in our experiments.

## 4 RELATED WORK

### 4.1 IMAGE RESTORATION

Image restoration aims to recover the latent clear images from their degraded version. With the development of the deep learning, convolution neural network (CNN)-based methods have dominated the image restoration tasks, and shows outstanding performance in dehazing (Chen et al., 2019; Liu et al., 2019; Li et al., 2020; Yu et al., 2022), deraining (Ren et al., 2019; Qian et al., 2018), deblurring (Gao et al., 2019; Ren et al., 2021; Cho et al., 2021), denoising (Abdelhamed et al., 2019; Chang et al., 2020) and many other tasks. Transformer-based methods (Liang et al., 2021; Zamir et al., 2022) have also shown impressive performance in image restoration.

### 4.2 DIFFUSION-BASED IMAGE RESTORATION

As diffusion models (Ho et al., 2020; Song et al., 2020a;b) have achieved unprecedented success in image generation (Dhariwal & Nichol, 2021) and various downstream tasks, such as image edition (Hertz et al., 2022; Brooks et al., 2023) and personalization (Ruiz et al., 2023; Gal et al., 2022), many works have explored its potential in image restoration tasks, which can be divided into two categories. The first one is to retrain diffusion models that are conditioned on the degraded images (Jiang et al., 2023; Yi et al., 2023; Saharia et al., 2022; Whang et al., 2022; Wei et al., 2023). Specifically, these methods diffuse the clear image into a pure Gaussian white noise in the forward process, and then recover the clear image by using the degraded image as the condition in the reverse process, which often lack interpretability. The second one is to adopt pre-trained diffusion models as prior to restore degraded images (Wang et al., 2023a; Feng et al., 2023; Mardani et al., 2023; Wang et al., 2023a; Fei et al., 2023; Wang et al., 2022). These methods do not require retraining the diffusion model, leading to relatively low computational overhead. Despite this, their performance is often limited. In this paper, we aim to propose a universal diffusion model for image restoration that can cover various degradation.

## 5 EXPERIMENTS

In this section, we conduct a comprehensive evaluation of the proposed R2Diff across multiple image restoration tasks, which encompass scenarios where $\mathbf{A} \neq \mathbf{E}$, such as low-light image enhancement, and those where $\mathbf{A} \approx \mathbf{E}$, such as deraining and denoising.

### 5.1 IMPLEMENTATION DETAILS

**Training Details.** Following the settings in (Ho et al., 2020), we use a U-Net as the denoising network for all experiments. The Adam optimizer is adopted to train the R2Diff with default settings. The learning rate is initialized to $1 \times 10^{-4}$, and decreases with a factor 0.5 every 200K iterations. The total iteration is set to 800K. The number of diffusion steps $T$ is set to 100 for all image restoration tasks. The mini-batch and the training patch size are set to 4, $256 \times 256$, respectively. We train our model on $2 \times 4090$Ti GPUs with PyTorch.

**Evaluation Metrics.** To evaluate the quality of the restored images, the commonly used Peak Signal to Noise Ratio (PSNR) and Structural Similarity Index Measure (SSIM) are adopted. Moreover, we use the Learned Perceptual Image Patch Similarity (LPIPS) and Fréchet inception distance (FID) to measure the perceptual difference.

Table 1: Quantitative comparison of different low-light image enhancement methods on LOL-v1 and LOL-v2 Real dataset. The best results and the second best ones are marked as **bold** and underline.

| Method | LOLv1 | | | | LOLv2-Real | | | |
|---|---|---|---|---|---|---|---|---|
| | PSNR ↑ | SSIM ↑ | LPIPS ↓ | FID ↓ | PSNR ↑ | SSIM ↑ | LPIPS ↓ | FID ↓ |
| RetinexNet | 16.77 | 0.462 | 0.417 | 126.27 | 17.72 | 0.652 | 0.436 | 133.91 |
| Zero-DCE | 14.86 | 0.562 | 0.372 | 87.24 | 18.06 | 0.580 | 0.352 | 80.45 |
| URetinex-Net | 19.97 | 0.828 | 0.267 | 62.38 | 21.13 | 0.827 | 0.208 | 49.84 |
| MPRNet | 20.46 | 0.772 | 0.293 | 75.62 | 22.37 | 0.832 | 0.296 | 63.23 |
| Restormer | 20.61 | 0.792 | 0.288 | 73.00 | 23.31 | 0.851 | 0.285 | 57.43 |
| WeatherDiff | 16.30 | 0.786 | 0.277 | 65.61 | 15.87 | 0.801 | 0.272 | 65.82 |
| ResShift | 19.23 | 0.735 | 0.225 | 61.21 | 20.41 | 0.704 | 0.218 | 60.72 |
| IR-SDE | 12.90 | 0.557 | 0.486 | 175.33 | 15.22 | 0.570 | 0.467 | 169.82 |
| GDP | 13.93 | 0.630 | 0.445 | 95.16 | 15.33 | 0.589 | 0.476 | 98.17 |
| R2Diff | **23.17** | **0.848** | **0.178** | **52.19** | **24.70** | **0.851** | **0.162** | **46.68** |

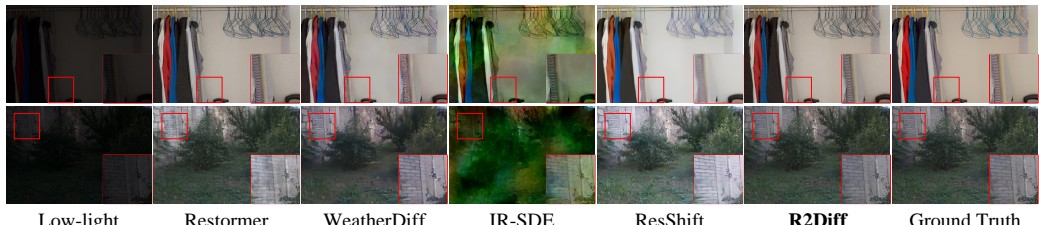

| Low-light | Restormer | WeatherDiff | IR-SDE | ResShift | **R2Diff** | Ground Truth |

Figure 3: Qualitative results of different low-light image enhancement methods on LOL datasets.

Table 2: Performance comparison of R2Diff on low-light image enhancement and deraining under different settings of $a$ and $p$. The best results are marked in **bold**.

| Coefficient | | LOLv1 | | | Coefficient | | Rain100L | | |
|---|---|---|---|---|---|---|---|---|---|
| $a$ | $p$ | PNSR ↑ | SSIM ↑ | LPIPS ↓ | $a$ | $p$ | PNSR ↑ | SSIM ↑ | LPIPS ↓ |
| | 0.1 | 21.32 | 0.796 | 0.387 | | 0.1 | 36.78 | 0.963 | 0.015 |
| | 0.3 | 21.56 | 0.805 | 0.262 | | 0.3 | 36.89 | 0.971 | 0.017 |
| 0.8 | 0.7 | 22.13 | 0.825 | 0.214 | 0.0 | 0.7 | **37.33** | **0.979** | **0.013** |
| | 3.0 | **23.17** | **0.848** | **0.178** | | 3.0 | 37.11 | 0.968 | 0.013 |
| | 5.0 | 22.60 | 0.836 | 0.193 | | 5.0 | 37.07 | 0.971 | 0.019 |
| 0.0 | | 20.11 | 0.763 | 0.387 | 0.0 | | **37.33** | **0.979** | **0.013** |
| 0.2 | | 20.67 | 0.786 | 0.326 | 0.2 | | 36.73 | 0.965 | 0.021 |
| 0.4 | 3.0 | 21.46 | 0.814 | 0.255 | 0.4 | 0.7 | 36.08 | 0.948 | 0.027 |
| 0.6 | | 22.34 | 0.827 | 0.241 | 0.6 | | 34.59 | 0.927 | 0.041 |
| 0.8 | | **23.17** | **0.848** | **0.178** | 0.8 | | 33.12 | 0.905 | 0.053 |
| 1.0 | | 23.01 | 0.841 | 0.183 | 1.0 | | 30.41 | 0.897 | 0.062 |

(a) Low-light image enhancement  (b) Deraining

## 5.2 EVALUATION ON LOW-LIGHT IMAGE ENHANCEMENT

**Comparison with state-of-the-art methods.** We first evaluate the R2Diff on low-light image enhancement, where **A** deviates from **E**. We choose the commonly-used LOL-v1 (Wei et al.) and LOL-v2 Real (Yang et al., 2021) as the evaluation datasets, which will be detailed in Appendix B.1. We compare the R2Diff with state-of-the-art CNN/Transformer-based methods, such as RetinexNet (Wei et al., 2018), Zero-DCE (Guo et al., 2020), URetinex-Net (Wu et al., 2022), MPRNet (Zamir et al., 2021), and Restormer (Zamir et al., 2022). We also compare with diffusion-based methods, such as GDP (Fei et al., 2023), WeatherDiff (Özdenizci & Legenstein, 2023), ResShift (Yue et al., 2023), and IR-SDE (Luo et al., 2023). As shown in Tab. 1, R2Diff surpasses all compared methods in all metrics. The higher PSNR and SSIM highlights the capability of R2Diff to yield image with better fidelity. And the lower scores in perceptual metric prove that the enhanced images by R2Diff can better serve for human visual system. Fig. 3 shows the qualitative results.

**Choice of $a$ and $p$ for low-light image enhancement.** The hyper-parameter $a$ in Eq. 8 governs the balance between ratio and residual, and $p$ in Eq. 16 controls the diffusion speed. It is crucial to make appropriate choices for $a$ and $p$ when handling different degradation. For low-light enhancement, the degradation operator **A** plays an important role in the degradation process (Wang et al., 2023b),

Table 3: Quantitative comparison of different image restoration methods on three image restoration methods, including deraining, deblurring, and denoising. The best results and the second best ones are marked as **bold** and underline. We mark N/A for those not applicable.

| Method | Deraining | | | Deblurring | | | Denoising | | |
|---|---|---|---|---|---|---|---|---|---|
| | PSNR↑ | SSIM↑ | LPIPS↓ | PSNR↑ | SSIM↑ | LPIPS↓ | PSNR↑ | SSIM↑ | LPIPS↓ |
| PReNet | 32.44 | 0.950 | 0.075 | N/A | N/A | N/A | N/A | N/A | N/A |
| MSPFN | 32.40 | 0.933 | 0.071 | N/A | N/A | N/A | N/A | N/A | N/A |
| DeepDeblur | N/A | N/A | N/A | 29.08 | 0.914 | 0.135 | N/A | N/A | N/A |
| DeblurGAN | N/A | N/A | N/A | 28.70 | 0.884 | 0.178 | N/A | N/A | N/A |
| DnCNN | N/A | N/A | N/A | N/A | N/A | N/A | 29.54 | 0.845 | 0.125 |
| BRDNet | N/A | N/A | N/A | N/A | N/A | N/A | 29.67 | 0.851 | 0.118 |
| MPRNet | 36.40 | 0.965 | 0.027 | 29.63 | 0.881 | 0.126 | **30.13** | **0.863** | 0.104 |
| AirNet | 34.90 | 0.966 | 0.058 | N/A | N/A | N/A | 30.02 | 0.855 | 0.113 |
| RainDiffusion | 36.85 | 0.972 | 0.036 | N/A | N/A | N/A | N/A | N/A | N/A |
| WeatherDiff | 35.27 | 0.968 | 0.021 | N/A | N/A | N/A | N/A | N/A | N/A |
| ResShift | 25.26 | 0.730 | 0.158 | 28.47 | 0.879 | 0.113 | 29.23 | 0.829 | 0.086 |
| IR-SDE | 36.51 | 0.979 | 0.016 | 30.21 | 0.892 | 0.107 | 29.41 | 0.836 | 0.071 |
| R2Diff | **37.33** | **0.979** | **0.013** | **30.44** | **0.896** | **0.068** | 29.46 | 0.839 | **0.068** |

(a) Deraining

(b) Deblurring

Figure 4: Qualitative results on different diffusion-based image restoration methods.

thus, the value of $a$ should be larger. On the other hand, dark areas typically occupy the whole low-light image, which means low-light significantly ruins the global content and color of images. Based on the analysis in Sec. 3.2, $p$ should be set larger, so that the diffusion can focus on enhancement when time-step $t$ is large. The comparison results in Tab. 2a provide supporting evidence for our statement. In our experiments, $a$ is set to 0.8, and $p$ is set to 3.0. As we can see, larger $a$ and $p$ tend to yield better restoration results. Note that $a = 1$ is not the best choice, since the noise term $\mathbf{n}$ also exists in the degradation process (Xu et al., 2022).

## 5.3 EVALUATION ON DERAINING

**Comparison with state-of-the-art methods.** We further test our method on deraining, where $\mathbf{A} \approx \mathbf{E}$ (Fu et al., 2021). The Rain100L (Yang et al., 2017) are utilized to evaluate different methods. We compare the R2Diff with PRENet (Ren et al., 2019), MSPFN (Jiang et al., 2020), MPRNet, AirNet (Li et al., 2022), RainDiffusion (Wei et al., 2023), WeatherDiff, and IR-SDE. As shown in Tab. 3, our method achieves the best performance in all three metrics compared with chosen baselines, which means that our method can recover clear image with higher fidelity (i.e., higher PSNR) and better perceptual quality (i.e., lower LPIPS). Fig. 4(a) shows the quanlitative results.

**Choice of $a$ and $p$ for deraining.** As illustrated in Fig 1(a), the degradation of rain can be directly represented through the residuals between clear and rainy images. So, for deraining, $a$ is set to 0.0. In fact, in this case, the R2Diff degenerates into a residual-based diffusion model. This highlights that we build a larger envelope space, which covers two specific modeling approaches: ratio-based

and residual-based. By adjusting $a$ flexibly, we can model various degradation processes effectively. As for $p$, we find that 0.7 is the best choice, as shown in Tab. 2b.

## 5.4 EVALUATION ON DEBLURRING, DENOISING, AND DEHAZING

We also evaluate the R2Diff on image deblurring and image denoising. For image deblurring, we choose GoPro (Nah et al., 2017) dataset for evaluation. For comparison, we compare the R2Diff with DeepDeblur (Nah et al., 2017), DeblurGAN (Kupyn et al., 2018), MPRNet, ResShift, and IR-SDE. In the experiments, $a$ is set to 0.1, and $p$ is set to 0.3, which will be discussed in Appendix B.2. As described in Tab. 3, the R2Diff outperforms all compared methods in terms of PSNR/SSIM. Furthermore, the R2Diff achieves the best LPIPS, which indicates that it can yield sharp images that look more realistic. We also provide qualitative results in Fig. 4.

For image denoising, we collect 3,950 clear images from Flickr2K (Timofte et al., 2017), DIV2K (Agustsson & Timofte, 2017), and BSD400 (Martin et al., 2001) for training. After that, we test our model on McMaster (Zhang et al., 2011) with noise level $\sigma = 50$. We compare our method with DnCNN (Zhang et al., 2017), BRDNet (Tian et al., 2020), AirNet, MPRNet, ResShift, and IR-SDE. $a$ is set to 0.0, and $p$ is set to 0.7, which will be detailed in Appendix B.2. As we can see from Tab. 3, the R2Diff can restore image with better perceptual quality (i.e., lower LPIPS), however, it falls shorts in terms of fidelity (i.e., lower PSNR and SSIM) compared with CNN-based methods. We further test our method on image dehazing, which is presented in Appendix B.3.

## 5.5 ABLATION STUDY

In this section, we perform comprehensive ablation studies to demonstrate the effectiveness of two components in the R2Diff.

**Effectiveness of Ratio and Residual Modeling.** A comprehensive analysis of the balance between ratio and residual for different degradation has been elaborately presented in Tab. 2. Taking low-light image enhancement as an example, as shown in Tab. 2a, it is evident that neither a diffusion model solely based on ratio ($a = 1.0$) nor one solely based on residual ($a = 0.0$) can accurately capture the intricacies of the low-light degradation. This is because low-light degradation includes not only the variation in illustration but also the presence of noise (Xu et al., 2022; Zheng et al., 2023). Hence, it is necessary to incorporate both ratio and residual in the diffusion process for effective low-light image enhancement.

**Effectiveness of the Exponential Schedule.** To flexibly control the diffusion speed for different degradation, we propose an exponential schedule for $\bar{\alpha}_t$ as formulated in Eq. 16, which is determined by a hyper-parameter $p$. To verify the effectiveness of our exponential schedule, we compare it with commonly-used linear and cosine schedules in existing diffusion probabilistic models (Ho et al., 2020) in low-light image enhancement on LOL-v1. Fig. 5 plot the PSNR learning curves of different schedules. As we can see, our schedule outperforms other schedules.

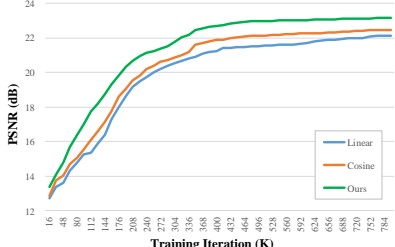

Figure 5: PSNR learning curves of different schedule.

## 6 CONCLUSION AND LIMITATION

In this paper, we propose a universal diffusion model for image restoration, called R2Diff. It consists of a Markov chain that transfer a high-quality image to its low-quality counterpart, whose transition kernel is constructed through the ratio and residual between them. An exponential schedule is further introduced to flexibly control the diffusion speed for different degradation. Extensive experiments on various image restoration tasks have demonstrated the superiority of our method.

On the other hand, there is still room for further improvement and refinement in our method. In our experiments, the hyper-parameters $a$ and $p$ for different degradation are manually tuned. In future research, we aim to explore an adaptive approach for adjusting the value of these hyper-parameters, which will eliminate the need for manual tuning, enhance efficiency, and potentially improve the performance of our method.

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

## A    DETAILED DERIVATION

### A.1    PROOF

In Eq. 8, we have $\mathbf{r}_0$. When $t = T$, we have:

$$\begin{aligned}
\mathbf{x}_0 + \mathbf{r}_0 &= \mathbf{x}_0 + a\left[\ln(\mathbf{y} + \mathbf{E}) - \ln(\mathbf{x}_0 + \mathbf{E})\right] + b(\mathbf{y} - \mathbf{x}_0) \\
&= \mathbf{x}_0 + \left[a\ln(\mathbf{y} + \mathbf{E}) + (1-a)\mathbf{y}\right] - \left[a\ln(\mathbf{x}_0 + \mathbf{E}) + (1-a)\mathbf{x}_0\right]
\end{aligned} \tag{17}$$

According to Taylor expansion, we have $\ln(\mathbf{x} + \mathbf{I}) = \mathbf{x} - \frac{\mathbf{x}^2}{2} + o(\mathbf{x}^2)$. Through removing the higher order terms, we can further get:

$$\begin{aligned}
\mathbf{x}_0 + \mathbf{r}_0 &= \mathbf{x}_0 + \left[a(\mathbf{y} - \frac{\mathbf{y}^2}{2} + o(\mathbf{y}^2)) + (1-a)\mathbf{y}\right] - \left[a(\mathbf{x}_0 - \frac{\mathbf{x}_0^2}{2} + o(\mathbf{x}_0^2)) + (1-a)\mathbf{x}_0\right] \\
&\approx \mathbf{x}_0 + (\mathbf{y} - \mathbf{x}_0) \\
&= \mathbf{y}
\end{aligned} \tag{18}$$

### A.2    DERIVATION OF EQ. 14

In the reverse process, we aim to recover $\mathbf{x}_{t-1}$ from $\mathbf{x}_t$. According to the Bayes's theorem, we can get:

$$q(\mathbf{x}_{t-1}|\mathbf{x}_t, \mathbf{x}_0, y) \propto q(\mathbf{x}_t|\mathbf{x}_{t-1}, y)q(\mathbf{x}_{t-1}|\mathbf{x}_0, y) \tag{19}$$

where $q(\mathbf{x}_t|\mathbf{x}_{t-1}, y) = \mathcal{N}(\mathbf{x}_t; \mathbf{x}_{t-1} + \alpha_t\mathbf{r}_0, \beta_t\mathbf{I})$ from Eq. 9, and $q(\mathbf{x}_{t-1}|\mathbf{x}_0, y) = \mathcal{N}(\mathbf{x}_{t-1}; \mathbf{x}_0 + \bar{\alpha}_{t-1}\mathbf{r}_0, \bar{\beta}_{t-1}\mathbf{I})$ from Eq. 10. Then, we focus on the exponential term,

$$\begin{aligned}
q(\mathbf{x}_{t-1}|\mathbf{x}_t, \mathbf{x}_0, \mathbf{y}) &\propto exp\{-\frac{(\mathbf{x}_t - \mathbf{x}_{t-1} - \alpha_t\mathbf{r}_0)^2}{2\beta_t} - \frac{(\mathbf{x}_{t-1} - \mathbf{x}_0 - \bar{\alpha}_{t-1}\mathbf{r}_0)}{2\bar{\beta}_{t-1}}\} \\
&= exp\{-\frac{\bar{\beta}_t}{2\beta_t\bar{\beta}_{t-1}}\mathbf{x}_{t-1}^2 + (\frac{\mathbf{x}_t - \alpha_t\mathbf{r}_0}{\beta_t} + \frac{\mathbf{x}_0 + \bar{\alpha}_{t-1}\mathbf{r}_0}{\bar{\beta}_{t-1}})\mathbf{x}_{t-1} + const.\} \\
&= exp\{\frac{(\mathbf{x}_{t-1} - \boldsymbol{\mu}_t(\mathbf{x}_t, \mathbf{x}_0, \mathbf{y}))^2}{2\boldsymbol{\sigma}_t^2} + const.\}
\end{aligned} \tag{20}$$

where the $const.$ is not related to $\mathbf{x}_{t-1}$, and

$$\begin{aligned}
\boldsymbol{\mu}_t(\mathbf{x}_t, \mathbf{x}_0, \mathbf{y}) &= \frac{\bar{\beta}_{t-1}}{\bar{\beta}_t}\mathbf{x}_t + \frac{\mathbf{K}_t}{\bar{\beta}_t}\left[a\ln(\mathbf{y}+\mathbf{E}) + b\mathbf{y}\right] + \frac{1}{\bar{\beta}_t}\left(\beta_t\mathbf{x}_0 - \mathbf{K}_t\left[a\ln(\mathbf{x}_0 + \mathbf{E}) + b\mathbf{x}_0\right]\right), \\
\boldsymbol{\sigma}_t^2 &= \frac{\beta_t\bar{\beta}_{t-1}}{\bar{\beta}_t},
\end{aligned} \tag{21}$$

In Eq. 21, we denote $\mathbf{K}_t = \beta_t\bar{\alpha}_{t-1} - \bar{\beta}_{t-1}\alpha_t$ for simplicity. Then, we can get the mean and variance of $q(\mathbf{x}_{t-1}|\mathbf{x}_t, \mathbf{x}_0, \mathbf{y})$ as represented in Eq. 13 and Eq. 14.

### A.3    DERIVATION OF EQ. 24

For the mean $\tilde{\boldsymbol{\mu}}_t(\mathbf{x}_t, \mathbf{x}_0, \mathbf{y})$ in Eq. 14, we can get $\boldsymbol{\mu}_\theta(\mathbf{x}_t, \mathbf{y}, t)$ through reparameterization:

$$\begin{aligned}
\boldsymbol{\mu}_\theta(\mathbf{x}_t, \mathbf{y}, t) &= \frac{\bar{\beta}_{t-1}}{\bar{\beta}_t}\mathbf{x}_t + \frac{\mathbf{K}_t}{\bar{\beta}_t}\left[a\ln(\mathbf{y} + \mathbf{E}) + b\mathbf{y}\right] \\
&+ \frac{1}{\bar{\beta}_t}(\beta_t f_\theta(\mathbf{x}_t, \mathbf{y}, t) - \mathbf{K}_t\left[a\ln(f_\theta(\mathbf{x}_t, \mathbf{y}, t) + \mathbf{E}) + bf_\theta(\mathbf{x}_t, \mathbf{y}, t)\right])
\end{aligned} \tag{22}$$

According to the loss function in Eq. 12, we obtain the simplified training objective:

$$L(\theta) = \sum_{t>0} \frac{1}{2\tilde{\beta}_t} \|\tilde{\boldsymbol{\mu}}_t(\mathbf{x}_t, \mathbf{x}_0, \mathbf{y}) - \boldsymbol{\mu}_\theta(\mathbf{x}_t, \mathbf{y}, t)\|_2^2 \qquad (23)$$

Since we have $\tilde{\boldsymbol{\mu}}_t(\mathbf{x}_t, \mathbf{x}_0, \mathbf{y})$ in Eq. 14 and $\boldsymbol{\mu}_\theta(\mathbf{x}_t, \mathbf{y}, t)$ in Eq. 23, we can further get:

$$L(\theta) = \sum_{t>0} \frac{1}{2\tilde{\beta}_t \bar{\beta}_t^2} \|\boldsymbol{\gamma}_t[f_\theta(\mathbf{x}_t, \mathbf{y}, t) - \mathbf{x}_0] - \boldsymbol{\lambda}_t[\ln(f_\theta(\mathbf{x}_t, \mathbf{y}, t) + \mathbf{E}) - \ln(\mathbf{x}_0 + \mathbf{E})]\|_2^2 \qquad (24)$$

where $\boldsymbol{\gamma}_t = \beta_t - \mathbf{K}_t b$ and $\boldsymbol{\lambda}_t = \mathbf{K}_t a$, which are dependent on time-step $t$ and hyper-parameter $a$ and $b$.

### A.4 FURTHER EXPLANATION OF EXPONENTIAL SCHEDULE

As mentioned in Sec. 3.2, for those degradation that primarily damages the global content of images (e.g., low-light), the $\bar{\alpha}_t$ should have a higher changing rate when time-step $t$ is large, as indicated by the green line in Fig. 6. Since the diffusion focuses on recovering the global content when $t$ is large, through setting $\bar{\alpha}_t$ with higher changing rate during that period, the diffusion model can fully perceive the degradation when $t$ is large. Consequently, the diffusion can effectively remove the degradation while recovering the global contents of images in the corresponding reverse process.

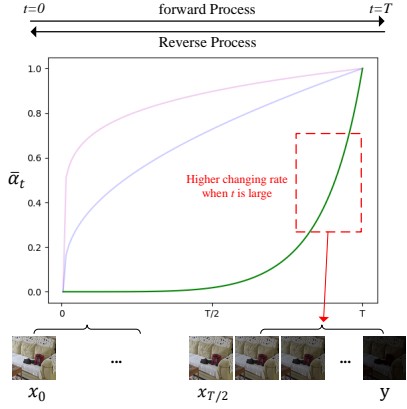

Figure 6: Choice of $\bar{\alpha}_t$.

## B EXPERIMENT

### B.1 DETAILS OF TRAINING DATASETS.

**Datasets for low-light image enhancement.** We evaluate our method for low-light image enhancement on LOL-v1 (Wei et al.) and LOL-v2-Real (Yang et al., 2021) dataset. The LOL-v1 contains 485 low-/normal-light image pairs for training, and 15 pairs for testing. The LOL-v2-Real includes 689 low-/normal-light image pairs for training, and 100 pairs for testing.

**Datasets for Deraining.** We evaluate our method for deraining on Rain100L (Yang et al., 2017), which consists of 200 rainy-clean image pairs for training, and 100 pairs for testing.

**Datasets for Deblurring.** We evaluate our method for deblurring on GoPro (Nah et al., 2017), which has 2,103 blurry-sharp image pairs for training, and 1,111 image pairs for testing. The blurry images in GoPro are generated by averaging the sharp images captured by a high-speed camera.

**Datasets for Denoising.** We use the combination of Flickr2K (Timofte et al., 2017), DIV2K (Agustsson & Timofte, 2017), and BSD400 (Martin et al., 2001) as training set for image denoising. Flickr2K includes 2,650 images, DIV2K has 1,000 images, and BSD400 contains 400 images.

### B.2 CHOICE OF $a$ AND $p$

**Choice of $a$ and $p$ for image deblurring.** For deblurring, we set $a$ and $p$ as 0.1 and 0.3, respectively. Tab. 4a shows the performance of R2Diff in image deblurring on GoPro under different settings of $a$ and $p$. As we can see, $p = 0.3$ is a better choice. Since deblurring requires recovering high-quality detailed textures from blurry images, $p$ should be set smaller, which encourages the diffusion model to restore detailed textures when $t$ is small. However, it may bring trade-off between fidelity and realism. As for $a$, we can see that $a = 0.1$ is slightly better than $a = 0.0$. Although the diffusion model based on residual only has proven its effectiveness in deblurring, the degradation operator $\mathbf{A}$

Table 4: Performance comparison of R2Diff on deblurring and denoising under different settings of $a$ and $p$. The best results are marked in **bold**.

| Coefficient | | GoPro | | | Coefficient | | McMaster | | |
|---|---|---|---|---|---|---|---|---|---|
| $a$ | $p$ | PNSR ↑ | SSIM ↑ | LPIPS ↓ | $a$ | $p$ | PNSR ↑ | SSIM ↑ | LPIPS ↓ |
| | 0.1 | 30.18 | 0.882 | 0.065 | | 0.1 | 28.63 | 0.814 | 0.086 |
| | 0.3 | **30.44** | **0.896** | 0.068 | | 0.3 | 28.58 | 0.826 | 0.073 |
| 0.1 | 0.7 | 30.30 | 0.887 | **0.053** | 0.0 | 0.7 | **29.46** | **0.839** | **0.068** |
| | 3.0 | 30.28 | 0.891 | 0.082 | | 3.0 | 29.23 | 0.832 | 0.078 |
| | 5.0 | 30.15 | 0.885 | 0.073 | | 5.0 | 28.97 | 0.828 | 0.075 |
| 0.0 | | 30.29 | 0.886 | 0.072 | 0.0 | | **29.46** | **0.839** | **0.068** |
| 0.1 | | **30.44** | **0.896** | **0.068** | 0.2 | | 28.45 | 0.805 | 0.123 |
| 0.4 | 0.3 | 30.05 | 0.879 | 0.075 | 0.4 | 0.7 | 26.13 | 0.763 | 0.288 |
| 0.6 | | 29.83 | 0.863 | 0.089 | 0.6 | | 22.28 | 0.816 | 0.463 |
| 0.8 | | 29.15 | 0.846 | 0.093 | 0.8 | | 20.75 | 0.688 | 0.572 |
| 1.0 | | 28.34 | 0.827 | 0.115 | 1.0 | | 18.33 | 0.561 | 0.661 |

|  (a) Deblurring | (b) Denoising |

plays an important role in deblurring (Fu et al., 2021; Nah et al., 2017), and should be modeled into the diffusion process.

**Choice of** $a$ **and** $p$ **for image denoising.** For image denoising, we set $a$ and $p$ as 0.0 and 0.7, respectively. Similar to image deraining, $\mathbf{A} = \mathbf{E}$ for image denoising. So we set $a = 0.0$, which means the R2Diff degenerates into a residual-based diffusion model. As we can see from Tab. 4b, increasing the value of $a$ will result in a decline in the denoising performance. As for $p$, we observe that $p = 0.7$ is the best choice.

### B.3  EVALUATION ON DEHAZING

We further test the R2Diff on image dehazing. We evaluate our method on NH-HAZE (Ancuti et al., 2020a;b), which contains 55 hazy-clean image pairs. In our experiments, $a$ and $p$ is set to 0.6 and 2.0. The qualitative results are provided in Fig. 7. As we can see, the R2Diff can remove the haze and recover contents of images.

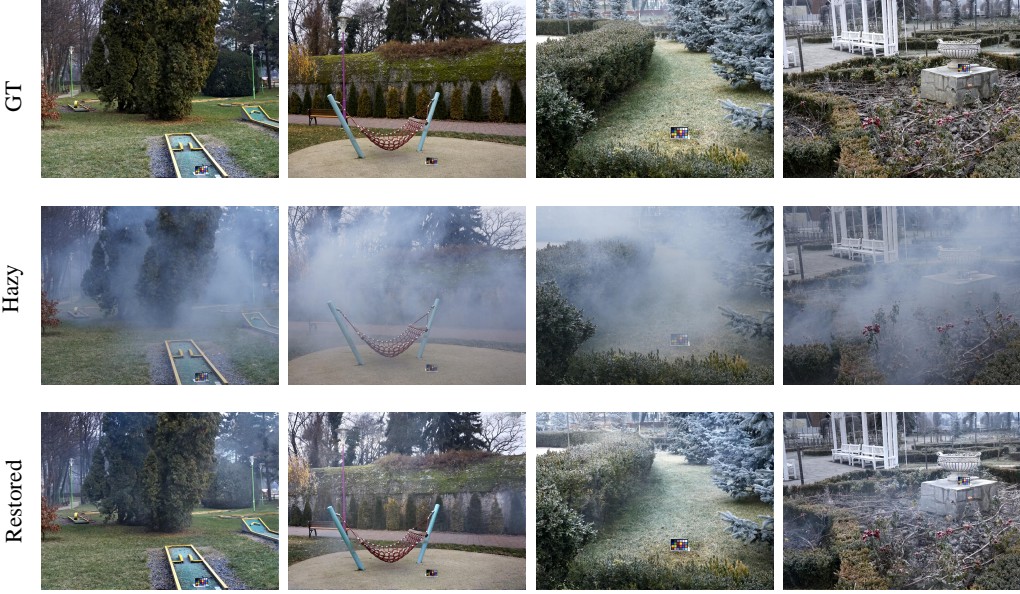

Figure 7: Qualitative results of dehazing on NH-HAZE dataset.

### B.4 MORE QUALITATIVE RESULTS

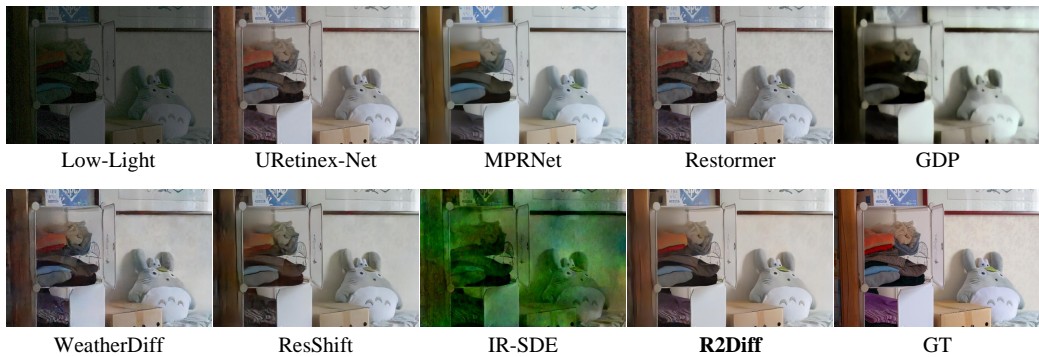

Low-Light    URetinex-Net    MPRNet    Restormer    GDP

WeatherDiff    ResShift    IR-SDE    **R2Diff**    GT

Figure 8: Qualitative results of low-light image enhancement on LOL-v1 dataset.

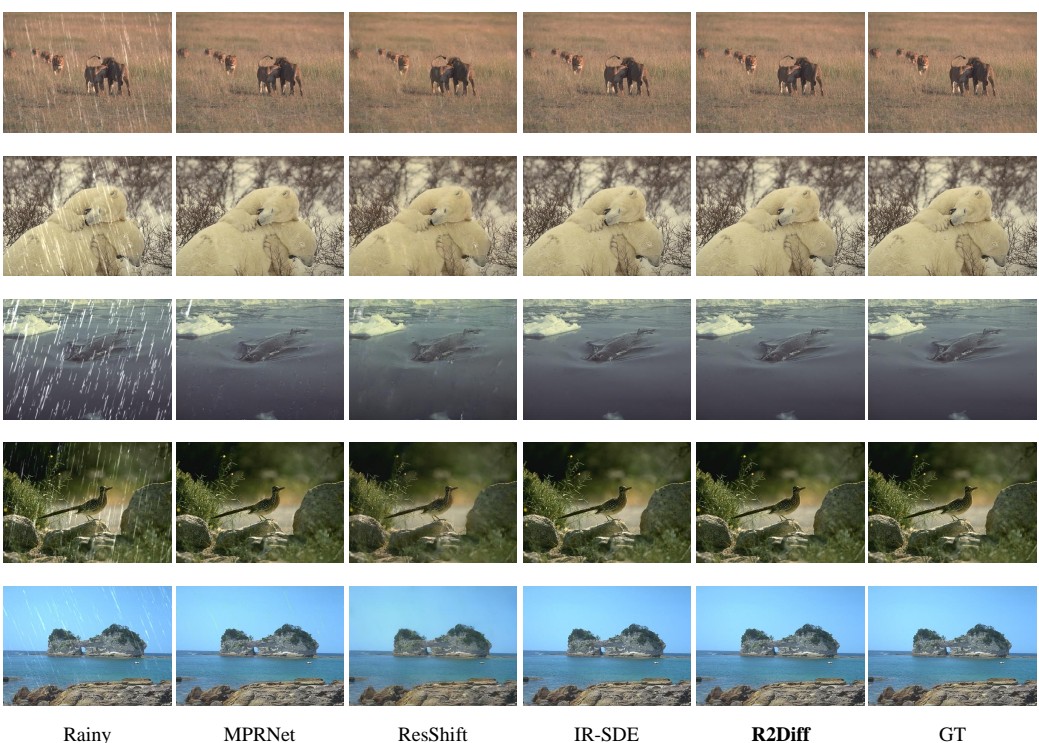

Rainy    MPRNet    ResShift    IR-SDE    **R2Diff**    GT

Figure 9: Qualitative results of deraining on Rain100L dataset.

