# OpenReview forum: "Ratio-Residual Diffusion Model for Image Restoration"
_ICLR.cc/2024/Conference — ICLR 2024 Conference Withdrawn Submission_

### Official Review · Reviewer_Cv16 · 2023-10-27

**Soundness:** 2 fair
**Presentation:** 2 fair
**Contribution:** 2 fair
**Rating:** 3
**Confidence:** 4

**Summary:**

This paper introduces a  diffusion model for image restoration that can cover different types of image degradation. The main idea is to formulate a Markov chain that converts a high-quality image to its low-quality counterpart. The transition kernel of this Markov chain is constructed through the ratio and residual between the high-quality and low-quality images, which provides a general expression that can effectively handle various degradation processes. Experiments show that the method outperforms existing image restoration methods and achieves superior performance on multiple image restoration tasks.

**Strengths:**

* The paper addresses the important and practical problem of image restoration with a diffusion strategy that is general to different degradations.
* Experimental results show that the method produces promising results on a variety of restoration benchmarks.

**Weaknesses:**

* The presentation and motivation of the method it's unclear
* Mathematical notation is not clear or wrongly defined
* Connections to previous work is unclear. In particular there is relevant published work that also propose to directly model the degradation from the high-quality to a low-quality image. See for example (and references therein):
  * Liu, G.H., Vahdat, A., Huang, D.A., Theodorou, E.A., Nie, W. and Anandkumar, A. I $^ 2$ SB: Image-to-Image Schr\" odinger Bridge. ICML 2023;
  * Delbracio, M, Milanfar, P. Inversion by Direct Iteration: An Alternative to Denoising Diffusion for Image Restoration, TMLR 2023;

**Questions:**

I would like the authors to comment on these major points.

1. The paper motivates the method through the classical linear degradation model: $y = Ax + n$. Then the idea of the residual is to model $(y-x) = (A-Id)x + n$, where Id is the Identity matrix.  But in the paper this is written as $(y-x) = (A-E)x + n$, with $E$ the "all-ones" matrix. This doesn't seem correct.

2. In the model, $y = Ax_0 + n$,  $A$ is a matrix (or linear operator). What does it mean to adopt $A = y /x_0$? This doesn't seem well defined. The same applies to $ln(y + E)$ in Eq(8).

3. Connection to published work. In particular,
  - In Delbracio and Milanfar, the diffusion (also known as a Bridge) is directly modeled as $x_t = (1-t)x + ty$. In the current linear formulation this leads to $x_t = ((1-t)Id + tA)x + tn$. This seems related to the formulation of the authors are proposing. I would like the authors to comment on the connection.

  - Also, I would like the authors to comment on the connection to Whang et al (2022), DvSR since in this work the idea is also to model a residual (but also colearn the initial base point of the degradation to with the residual is computed).

---

### Official Review · Reviewer_RMrE · 2023-10-30

**Soundness:** 3 good
**Presentation:** 3 good
**Contribution:** 2 fair
**Rating:** 5
**Confidence:** 2

**Summary:**

This paper presents a diffusion model that uses a Markov chain to transform high-quality images into low-quality ones. This Markov chain's transition kernel is derived from the ratio and residual between the two image qualities, offering an effective approach to manage different degradation processes. We design an exponential schedule that allows flexible control over the diffusion speed for diverse degradation, resulting in enhanced restoration performance. Extensive experiments demonstrate our method's superiority or comparability to existing image restoration techniques across multiple tasks, such as low-light image enhancement, deraining, deblurring, denoising, and dehazing.

**Strengths:**

1. The performance is excellent. The experiments are conducted on extensive datasets for several tasks. The proposed method competes with all previous state-of-the-art methods.
2. The paper is clearly written. The main ideas are conveyed clearly and can be easily understood.

**Weaknesses:**

1. In general, the proposed ideas are relatively straightforward, and it is not easy to directly understand the real significant novelties and contributions of the paper.
2. The discussion of existing work is insufficient.
The following highly correlated works should be discussed:
[1] Zehua Chen, Yihan Wu, Yichong Leng, Jiawei Chen, Haohe Liu, Xu Tan, Yang Cui, Ke Wang, Lei He, Sheng Zhao, Jiang Bian, Danilo Mandic, "ResGrad: Residual Denoising Diffusion Probabilistic Models for Text to Speech," arXiv, 2022.
[2] Yi Zhang, Xiaoyu Shi, Dasong Li, Xiaogang Wang, Jian Wang, Hongsheng Li, "A Unified Conditional Framework for Diffusion-based Image Restoration," arXiv, 2023.
[3] Jiawei Liu, Qiang Wang, Huijie Fan, Yinong Wang, Yandong Tang, Liangqiong Qu, "Residual Denoising Diffusion Models," arXiv, 2023.
[4] Noor Fathima Ghouse, Jens Petersen, Auke Wiggers, Tianlin Xu, Guillaume Sautiere, "A Residual Diffusion Model for High Perceptual Quality Codec Augmentation," arXiv, 2023.

The reviewer understands that it may be considered somewhat challenging to request the authors to cite and discuss arXiv papers. However, the diffusion-based methods are too fast evolving. It is beneficial for the reviewers and authors to see those discussions to distinguish the real merits of the paper compared to these different technical routes.

3. The ablation studies are lacking. It is hard to see where the performance gains come from.
4. Some results are not provided in a convincing way. For example, in Table I, Restormer achieves much better results than other restormer baselines. Are all methods retrained on the same datasets following the same protocal?
5. Besides, the FID results make no sense in restoration tasks and are suggested to be removed.

**Questions:**

Please see the weakness part.

---

### Official Review · Reviewer_xayc · 2023-11-01

**Soundness:** 3 good
**Presentation:** 3 good
**Contribution:** 3 good
**Rating:** 8
**Confidence:** 4

**Summary:**

The paper proposed a framework for general image restoration tasks using diffusion model. The logarithmic representation of degradation operator and residual representation of noise operator are adopted and combined as transition from clean to degraded image. An exponential schedule is designed to control the changing rate of diffusion process.

**Strengths:**

The proposed transition takes both degradation operator and noise operator into consideration which is a more general and accurate modelling of image degradation.

The designed scheduler provides flexibility for controlling the whole process when dealing with different image degradations.

**Weaknesses:**

There is no universal setting for different degradations. You will need to manually set the weighting factor of degradation operator and noise as well as the scheduler to get the best results for different tasks. Judging from the experiment results in Table 2, the output of the model also seems to be quite sensitive to these settings.

**Questions:**

Any idea of designing automated parameter searching algorithms?

Would be interesting to see some examples of the proposed method working on mixed degradation case (for example, noise+blur).